# Identification of Cardiovascular Disease-Related Genes Based on the Co-Expression Network Analysis of Genome-Wide Blood Transcriptome

**DOI:** 10.3390/cells11182867

**Published:** 2022-09-14

**Authors:** Taesic Lee, Sangwon Hwang, Dong Min Seo, Ha Chul Shin, Hyun Soo Kim, Jang-Young Kim, Young Uh

**Affiliations:** 1Division of Data Mining and Computational Biology, Institute of Global Health Care and Development, Wonju Severance Christian Hospital, Wonju 26411, Korea; 2Department of Family Medicine, Yonsei University Wonju College of Medicine, Wonju 26411, Korea; 3Artificial Intelligence Bigdata Medical Center, Yonsei University Wonju College of Medicine, Wonju 26411, Korea; 4Department of Medical Information, Yonsei University Wonju College of Medicine, Wonju 26411, Korea; 5Pharmicell Co., Ltd., Seongnam 13229, Korea; 6Department of Internal Medicine, Yonsei University Wonju College of Medicine, Wonju 26411, Korea; 7Department of Laboratory Medicine, Yonsei University Wonju College of Medicine, Wonju 26411, Korea

**Keywords:** cardiovascular disease-related transcriptomic signature, cardiovascular disease-related gene, differential expression, differential co-expression, disease-related modules

## Abstract

Inference of co-expression network and identification of disease-related modules and gene sets can help us understand disease-related molecular pathophysiology. We aimed to identify a cardiovascular disease (CVD)-related transcriptomic signature, specifically, in peripheral blood tissue, based on differential expression (DE) and differential co-expression (DcoE) analyses. Publicly available blood sample datasets for coronary artery disease (CAD) and acute coronary syndrome (ACS) statuses were integrated to establish a co-expression network. A weighted gene co-expression network analysis was used to construct modules that include genes with highly correlated expression values. The DE criterion is a linear regression with module eigengenes for module-specific genes calculated from principal component analysis and disease status as the dependent and independent variables, respectively. The DcoE criterion is a paired *t*-test for intramodular connectivity between disease and matched control statuses. A total of 21 and 23 modules were established from CAD status- and ACS-related datasets, respectively, of which six modules per disease status (i.e., obstructive CAD and ACS) were selected based on the DE and DcoE criteria. For each module, gene–gene interactions with extremely high correlation coefficients were individually selected under the two conditions. Genes displaying a significant change in the number of edges (gene–gene interaction) were selected. A total of 6, 10, and 7 genes in each of the three modules were identified as potential CAD status-related genes, and 14 and 8 genes in each of the two modules were selected as ACS-related genes. Our study identified gene sets and genes that were dysregulated in CVD blood samples. These findings may contribute to the understanding of CVD pathophysiology.

## 1. Introduction

Cardiovascular disease (CVD) is a major cause of premature mortality that contributes to disability [1]. Representative modifiable risk factors from accumulating clinical evidence include high systolic blood pressure, fasting plasma glucose, and low-density lipoprotein cholesterol [1]. Traditional prevention and treatment strategies for the cardiometabolic risk factors [1] can effectively reduce the risk of atherosclerotic CVD [2]. Systems genetics, a new and promising strategy, has increased our global understanding of the flow of biological information underlying complex traits, phenotypes, and diseases [3,4].

In the last decade, genome-wide association studies (GWASs) have played a pioneering role in systems genetics research. For coronary artery disease (CAD) alone, GWASs have revealed more than 150 CAD-related single-nucleotide polymorphisms (SNPs) [5,6]. However, most of the identified phenotype-, trait-, or disease-related loci are not located in protein-coding regions [7]. This makes it difficult to trace or interpret their effects on downstream genes. Moreover, the genetic determinants of complex traits, phenotypes, or diseases cannot be explained using GWAS data alone. This limitation is not exclusive to GWAS, as no mono-omic approach (such as transcriptome, methylome, or proteome) can explain all the genetic signatures of complex traits [8]. Therefore, integrative analyses of other omics, such as transcriptome and methylome, have been performed to expand our understanding of CVD-related molecular signatures [9,10,11,12].

The Stockholm Atherosclerosis Gene Expression (STAGE) study involved from 2009 performed whole-genome transcriptome analyses in five tissues and identified an atherosclerosis module that included LIM domain-binding protein 2 (LDB2) as a high-hierarchy regulator [9]. In 2016, the STAGE also collected genetic and transcriptomic data from seven tissues and conducted integrative analyses of multi-tissue co-expression networks, expression quantitative trait loci (eQTLs), and GWAS, and constructed 30 intercorrelated CAD-related regulatory gene networks (RGNs) among vascular and metabolic tissues [10]. The STARNET study included a genome-wide transcriptome analysis of seven tissues and replicated 28 RGNs proposed by the STAGE [10], which showed strong associations with CVD variations in humans and mice [13].

Peripheral blood is readily available from both humans and mice, and recent evidence has shown that it can reflect the transcriptomic signature of other tissues [14,15]. Lee and Lee [16] utilized blood transcriptome data and identified several CVD-related gene sets using statistical methods, disease-gene databases, and eQTL and GWAS summary statistics. However, only differential expression (DE) analysis was conducted in this study [16], and, therefore, it could not explain the co-dysregulated patterns among disease status genes. Several studies have performed differential co-expression (DcoE) analyses to identify altered gene–gene interactions under disease conditions [17,18]. In this study, we integrated DE and DcoE analyses to identify CVD-related gene sets (also termed modules). Specifically, we categorized four blood CVD datasets into obstructive CAD status- and ACS-related datasets and separately identified the obstructive CAD- and ACS-related gene sets based on co-expression network analysis. Additional statistical and mathematical analyses (i.e., gene–gene interaction networks) were used to identify potential candidate genes involved in CVD pathogenesis or progression. Finally, a replication study using independent human blood and multi-tissue mouse datasets was performed to identify robust CVD-related genes.

## 2. Materials and Methods

The four main tasks conducted in this study were as follows: first, a large dataset was constructed by integrating blood transcriptome datasets via selecting and removing batch effects among them (Figure 1). Second, modules, including genes exhibiting similar expression patterns, were established using co-expression network analysis. Third, the module information was applied to each blood dataset, not batch-normalized, and modules related to the CVD status were selected based on the DE and DcoE criteria. Fourth, we finalized the disease-related modules or genes common among the two blood datasets (GSE90074-GSE20681 or GSE34198-GSE60993). All tasks, including integrating datasets, establishing a co-expression network, and selecting CVD, were conducted using R language (version 4.0.1, R Foundation for Statistical Computing, Vienna, Austria), and the source code is available at our research group site (https://github.com/WCH-AI-LAB/, accessed on 9 September 2022).

### 2.1. Datasets

A previous study retrieved 11 blood CVD gene expression datasets [16] from the Gene Expression Omnibus (GEO) database [19]. The integration of numerous transcriptomic datasets can cause several problems, including the loss of many transcripts (also termed as probe or probe-set) and disease-related signatures (such as fold change (FC) between two statuses) [18]. Therefore, Lee and Lee [16] selected the blood CVD gene expression datasets for the final analysis because of the good quality of these datasets, as measured by MetaQC tool [20]. In this study, four blood datasets were selected as representative CVD datasets.

The GSE90074 dataset included 93 obstructive CAD and 50 non-obstructive CAD blood samples [21]. Blood samples were obtained from patients enrolled in the supporting multidisciplinary approach to research atherosclerosis (SAMARA) study [21]. The GSE20681 dataset consisted of 99 CAD blood samples with more than 50% stenosis in one or more major coronary vessels and 99 matched controls (CNs) with <50% stenosis in all major vessels [22,23]. These blood samples (GSE20681) were obtained from PREDICT, a multicenter study performed in the United States, comprising patients referred for coronary angiography. The GSE34198 dataset comprised 45 patients with acute myocardial infarction (AMI) and 48 matched controls [24]. The GSE60993 dataset included information obtained from the blood samples of 26 patients with acute coronary syndrome (ACS), who underwent coronary angiography or primary percutaneous coronary intervention at Seoul St. Mary’s Hospital in South Korea, and seven individuals with normal coronary angiography (CN) [25]. The overall information for the transcriptomic datasets analyzed in the present study is summarized in the Appendix A).

### 2.2. Preprocessing

The RNA expression values of the GSE90074 and GSE20681 datasets were subjected to quantile normalization. Expression values for mRNA in GSE34198 were scaled and normalized using log-transformation and quantile methods, respectively. The gene expression in GSE60993 was previously subjected to quantile normalization.

Probes or probe sets presenting minor variances were excluded. We calculated the standard deviations (SDs) of the expression levels for all genes across samples and then removed the genes with low SDs of 40% (Appendix A). GSE90074, GSE20681, GSE34198, and GSE60993 datasets measured RNA expression using Agilent and Illumina platforms, respectively. Therefore, they had different transcript identifiers (IDs) for the probes or probe sets. To integrate the two blood datasets into a large dataset, all probe or probe set IDs were remapped to the Entrez IDs. For multiple probes or probe sets that were annotated with a gene-based Entrez ID, the probe with the maximum mean expression value using the “collapseRows” function in the weighted gene co-expression network analysis (WGCNA) package [26] was used.

### 2.3. DE Analysis

DE analysis between the two statuses (CAD with high stenosis vs. CAD with low stenosis and ACS vs. CN) was performed using the limma package [27]. In the limma method, “lmFit” and “eBayes” functions were used to conduct the DE analysis. Summary statistics from the limma method consisted of gene-based FC values between two statuses and their *t*- and *p*-values. Genes with a false discovery rate (FDR)-adjusted *p* < 0.05 were defined as the differentially expressed genes (DEGs). Genes exhibiting uncorrected *p* < 0.05 were defined as the possible DEGs.

### 2.4. Construction of Co-Expression Network and Modules

To systematically capture gene–gene interactions based on the whole transcriptome and their differential signatures between disease and control states, we established modules consisting of genes with similar gene expression patterns using the WGCNA package [26]. Raw and large gene expression datasets were processed to eliminate the batch effects between the two datasets (GSE90074-GSE20681 or GSE34198-GSE60993) using the ComBat method [28,29]. For the batch removal between two different datasets, the “ComBat” function in the sva package [28] was used.

The construction of a co-expression network began with the establishment of a pairwise correlation matrix between all possible genes. Biweight midcorrelation was used to establish the correlation matrix [18,30]. The biweight midcorrelation is a median-based correlation estimate and is, therefore, robust to outliers. The correlation matrix was converted into an adjacency matrix by using a β-squared function. The parameter β was determined when an approximate scale-free topology was achieved (R^2^ > 0.8). Then, the topological overlap matrix (TOM) was calculated from the adjacency matrix to reduce noise by setting “signed” as “TOMType” [26]. A network dendrogram was constructed using average-based hierarchical clustering for the dissimilarity matrix (1 − TOM). The modules were determined by applying the top-down dynamic tree-cut method to the hierarchical dendrogram. For the construction of the module, “adjacency” and “cutreeHybrid” functions in the WGCNA package [26] were used.

### 2.5. Selection of CVD-Related Modules

Two main methods are used to characterize or select disease-related modules. Gandal et al. [30] selected several modules and characterized them based on the DE analysis between the control and disease status (Appendix A). Zhang et al. [17] identified modules exhibiting significant correlational differences among genes in a specific module between two statuses. Considering these criteria, we selected the modules satisfying both the DE and DcoE criteria as in a previous study [18]. As the initial step of the DE analysis, we calculated the first principal component (module eigengene, ME) of a given module (Appendix A). Linear regression was used to identify the association between disease-related modules based on the DE analysis, with the ME and disease status set as the dependent and independent variables, respectively.

For the DcoE analysis, we constructed four adjacency matrices from two datasets and two disease statuses. The intramodular connectivity among all the genes in a module was calculated for the six matrices using the “intramodularConnectivity” function in the WGCNA package [26]. For example, if a module consists of 500 genes, 500 values of intramodular connectivity are separately calculated for both the control and disease statuses. Then, the intramodular connectivity between the disease and control groups for each dataset was compared using a paired *t*-test. A module with a Bonferroni-corrected *p* < 0.05 was selected as the disease-related module based on the DcoE analysis.

### 2.6. Identification of CVD-Related Genes

Several machine-learning-based network construction methods have been used to identify hub or upregulated genes [9,16,18]. However, they poorly explain (low interpretability or explainability) the inferred edges or interactions between the two genes. Therefore, we implemented mathematical (e.g., calculation of the correlation matrix) and statistical methods (e.g., selection of extreme values based on Z-statistics) to improve the interpretability of the estimated gene–gene interaction.

To select genes with significantly increased or decreased connectivity in the disease status, we constructed disease- and control-dissimilarity matrices (1 − TOM) using CVD and matched CN samples, respectively. For each matrix, gene–gene pairs with less than average minus three SDs of dissimilarity values were regarded as the presence of an interaction. We then calculated the number of gene interactions and compared the differences between the edges of the genes in the disease and matched controls. Based on Z-scores of ±2, genes with increased or decreased interactions with other genes in a disease module compared with those in the matched CN module were selected.

### 2.7. Pathway Analysis

The degree of similarity between candidate gene sets curated from our study and known gene sets obtained from the pathway database was measured using a hypergeometric test as follows: $$p-\mathrm{value}\;(\mathrm{hypergeometric}\;\mathrm{test})=\overset{\min\left(c,\;p\right)}{\underset{k=i}{\sum }}\frac{\left(\begin{array}{c} c\\ k \end{array}\right)\left(\begin{array}{c} N-c\\ p-k \end{array}\right)}{\left(\begin{array}{c} N\\ p \end{array}\right)},$$
where *N* represents the total number of genes (referred to as the number of background genes) in the gene expression dataset, *p* represents the number of genes in a known gene set, *c* represents the number of genes in a candidate gene set, and *i* represents the number of genes common between the candidate gene set and known gene set. 

Known gene sets (called pathways) with an FDR-corrected *p* < 0.05 were determined as significantly enriched pathways. The Kyoto Encyclopedia of Genes and Genomes (KEGG) [31] and Gene Ontology (GO) [32] databases were used as the pathway databases, which were downloaded from the Molecular Signature Database (MSigDB) [33].

### 2.8. Validation of the Candidate Genes

Validation studies for candidate genes obtained from transcriptome data are typically conducted using several methods, such as in vitro and in vivo experiments, quantitative real-time polymerase chain reaction (real time-qPCR), and replication studies using an independent dataset. Through statistical analyses, Soh et al. [34] identified cancer-related miRNAs associated with somatic copy number alterations and validated them in vitro by measuring the viability and proliferation rates of cells transfected with their inhibitors. Oh et al. [35] identified aging-related genes in the kidney tissue using a regression method and validated them in vivo using mouse and zebrafish models. Joehanes et al. [36] selected genes related to coronary heart disease based on DE analysis and conducted a validation study using real time-qPCR. Lee and Lee [18] conducted a replication study using independent public datasets to validate the candidate genes. Among the aforementioned validation methods, we conducted a replication study using independent human and mouse datasets.

For human transcriptome datasets independent of GSE90074, GSE20681, GSE34198, and GSE60993, we selected the blood CVD dataset GSE59867. For mouse gene expression datasets used to validate candidate genes, the heart (GSE4648, GSE49937, GSE153485, and GSE775), liver (GSE153485), muscle (GSE153485), and white adipose tissue (WAT, GSE153485) CVD datasets were selected. All probe or probe set IDs of the human and mouse datasets (external validation sets) were remapped to the human Entrez IDs. For multiple probes or probe sets matched with an Entrez ID, the probe with the maximum average expression value is selected as described in Section 2.2.

## 3. Results

### 3.1. Comparisons of Disease-Related Signatures among the Four Blood CVD Datasets

Obstructive CAD- or ACS-related signatures (i.e., FCs between the two statuses for all genes) were compared based on Pearson’s and Spearman’s correlation coefficients (PCC and SCC). A blood dataset (GSE90074) exhibited a high correlation between CAD obstruction-related signatures and those from GSE20681 (PCC:0.307, SCC:0.339, Figure 2A). The highest correlation coefficient was obtained in the comparison of disease-related signatures between GSE20681 and GSE60993 (Figure 2A). Two comparative pairs (GSE90074–GSE20681 and GSE34198–GSE60993) exhibited significantly high correlations (Figure 2A).

In four derivation datasets (GSE90074, GSE20681, GSE34198, and GSE60993), no DEG satisfying multiple comparison tests were identified. Therefore, based on uncorrected *p*-values, possible DEGs between the two conditions were identified (Figure 2B). GSE90074, the blood CAD obstruction dataset, provided 689 possible DEGs between CAD with high and low stenosis (Figure 2B). The 689 DEGs from GSE90074 significantly overlapped with those from GSE20681 and GSE60993 based on the hypergeometric test. Most of the comparative pairs among the possible DEGs in the four blood datasets were significantly correlated (Figure 2B). Most pathways enriched by the four lists of possible DEGs from the four blood datasets were immune- and inflammation-related KEGG or GO pathways (Figure 2C).

### 3.2. Establishment of Modules

Four blood CVD datasets (GSE90074, GSE20681, GSE34198, and GSE60993) were categorized into obstructive CAD- and ACS-related datasets and then separately integrated into two large blood datasets (Figure 1). Only the genes present in each of the two datasets were retained for subsequent analyses. The overall distribution of expression for all genes did not vary across different phenotypes (e.g., obstructive CAD vs. non-obstructive CAD) but exhibited differential patterns among different platforms for measuring gene expression (Appendix A). In other words, the distribution of the whole gene expression was intact in the obstructive CAD or ACS compared with that in matched controls and was mainly determined by microarray platforms (Appendix A). Therefore, we normalized the batch effects between the two datasets (GSE90074–GSE20681 and GSE34198–GSE60993) using the ComBat method (Appendix A).

For the combined blood CAD status (high vs. low stenosis) gene expression dataset (GSE90074 and GSE20681), a correlation matrix was calculated using the biweight midcorrelation method. Then, the 12 power values satisfying the scale-free topology *R^2^* fitting index > 0.8 (Appendix A) were selected to convert the correlation matrix to an adjacency matrix. For the ACS dataset (GSE34198 and GSE60993), the power value was set to 15 (Appendix A). From the adjacency matrix, the TOM and dissimilarity matrices were subsequently constructed.

Four parameters (partitioning around medoids (PAM), minimal module size, deep split, and maximum dissimilarity) were used to establish the module. In a previous study, we found that all genes tended to be excessively assigned to more than one module when the PAM stage was set on [18]. Therefore, the PAM stage was set off to eliminate the genes that were not arranged in any cluster. We then obtained 60 cases from six cases of minimal module size (50, 60, 70, 80, 90, and 100), five types of deep split (0, 1, 2, 3, and 4), and two cases of maximum dissimilarity (0.1 and 0.2). Sixty iterations were performed wherein modules were created based on the cases (Appendix A). The minimal module size, deep split, and maximum dissimilarity parameter values of 80, 3, and 0.1, respectively, were used to construct robust modules that retained more than half of the 60 iterations (Appendix A), yielding 21 modules in GSE90074–GSE20681 (Figure 3A). For the large dataset obtained from GSE34198 and GSE60094 (Appendix A), the minimal module size, deep split, and maximum dissimilarity parameter values were set to 100, 2, and 0.2, respectively, resulting in 23 gene sets (Figure 3B).

### 3.3. Identification of CVD-Related Module

Module selection or identification of disease-related genes is typically performed using integrated datasets [16,18,30,37]. Our study analyzed two blood CVD datasets, including different comparative combinations of phenotypes, such as CAD-low and high stenosis and ACS-CN. Lee and Lee [18] suggested that a significant amount of information in individual datasets may be lost if the datasets are intuitively integrated without a special strategy. Therefore, the original datasets that were not batch-removed were processed to identify disease-related modules or genes (Figure 1).

Among the 21 modules established by GSE90074 and GSE20681, we selected CAD status-related modules based on the DE analysis. For this analysis, we measured MEs, which are the first principal components of all genes in a module (Appendix A). The association between MEs and phenotypes, including obstructive and non-obstructive CAD, was tested using linear regression (Appendix A). Six modules exhibited significant obstructive CAD-related alterations compared with non-obstructive CAD in one or more of the two blood datasets (Figure 4A and Appendix A). Therefore, six modules were selected in the first step of CAD obstruction status-related module selection (Appendix A).

For the 23 modules constructed using GSE34198 and GSE60993, MEs summarized from six modules exhibited ACS-related dysregulation in one or more of the two datasets (Figure 4B and Appendix A). Based on the DE criterion, six modules (black, pink, brown, midnightblue, dark red, and gray60) were identified (Figure 4B). Note that the black module related to the CAD status included genes different from those in the black module curated from the ACS datasets.

In the second module selection step, DcoE analysis was applied to each of the six modules (Appendix A). First, the black, magenta, and brown modules were selected because their MEs significantly increased in obstructive CAD (CAD with high stenosis) compared with those in the non-obstructive CAD (CAD with low stenosis) in the two blood datasets (Figure 5). In other words, the overall expression of the genes in the black, magenta, and brown modules tended to be upregulated in CAD with high stenosis (Figure 5A). Correlations between genes in the black, magenta, and brown modules were higher in the obstructive CAD group than that in the matched control status group in the two blood datasets (Figure 5A and Appendix AA). Collectively, genes in the black, magenta, and brown modules were selected because they exhibited increased expression and gain of connectivity (GOC) in CAD with high stenosis compared with that in the matched control status (CAD with low stenosis). 

In case of ACS-related modules, the pink module was identified because the constituent genes were downregulated in one of the two blood datasets (GSE34198 and GSE60993) and exhibited loss of connectivity (LOC) in the two blood datasets in the ACS group (Figure 5B and Appendix AB). The dark-red module was selected because it exhibited upregulation and GOC in ACS group.

The magenta module contained 246 genes (Appendix A). Using hypergeometric tests and pathway databases such as the KEGG and GO databases, the biological function of turquoise-specific genes was identified. The results suggested that the proteasomal protein catabolic process (GO), vesicle organization (GO), proteasome-mediated ubiquitin-dependent protein catabolic process (GO), and cellular response to oxygen levels (GO) were enriched in the black-involved genes. Other obstructive CAD-related modules, such as black and brown gene sets, contained 267 and 396 genes, respectively (Appendix A).

Among the 246 genes in the magenta module, 16 exhibited an increased number in interactions with Z-scores ≥ 1 in the obstructive CAD status compared with that in matched controls for one or more of the two datasets. Of the 16 genes, 10 (*ATP6V1A*, *BASP1*, *CHMP2A*, *GCA*, *HNRNPH2*, *HSD17B11*, *NRDC*, *SLC16A3*, *TKT*, and *ZNF281*) exhibited positive FC values in both blood datasets (Figure 6B). Based on the same criteria with the method selecting disease-related genes in the magenta module, six genes (*COG3, CUTC, MAML1, MORF4L1*, *NPTN*, and *VTA1*) and seven genes (*ACSL1, AGO4, CEBPB, JPT1, RAB5IF, RNF130*, and *TALDO1*) were identified in black and brown modules, respectively (Figure 6A–C).

Among the 373 genes in the pink module (Appendix A), 14 genes (*CBX6*, *ERI3*, *FAM50A*, *FIBP*, *GNL1*, *HGH1*, *PARP6*, *PEX26*, *PHGDH*, *POU2F2*, *SNRPB*, *SPHK2*, *YIF1A*, and *ZNF296*) exhibited decreased expression (both datasets) and loss of connectivity patterns (one or more of the two datasets) in the ACS status (Figure 7A). The dark-red module included 265 genes (Appendix A), among which eight genes (*ALOX5AP, APMAP, B4GALT5, CHST15, HAL, LBR, SLC22A15*, and *STX3*) showed increased expression and gain of connectivity in the ACS status (Figure 7B).

### 3.4. Validation for the Candidate Genes

To identify robust differential expression and co-expression signatures in blood CVD datasets (GSE90074, GSE20681, GSE34198, and GSE60993), we conducted a replication study using an independent human blood dataset (GSE59867). GSE59867 includes two groups, ACS and stable CAD patients, as the phenotypes of interest and matched controls, respectively. The human blood CVD dataset GSE59867 contains gene expression values for 243 of the 246 genes in the magenta module. For all possible pairs of magenta-specific genes (243 genes), we constructed two separate Spearman correlation matrices for ACS and stable CAD samples per dataset. Among the 10 genes, nine (*ATP6V1A*, *BASP1*, *CHMP2A*, *GCA*, *HNRNPH2*, *HSD17B11*, *NRDC*, *TKT*, and *ZNF281*) exhibited GOC in ACS group of the GSE59867 (Figure 8). Among the nine genes, five (*ATP6V1A*, *BASP1*, *CHMP2A*, *GCA*, and *TKT*) were dysregulated in the ACS status (Figure 8).

In the brown module, all seven genes (*ACSL1*, *AGO4*, *CEBPB*, *JPT1*, *RAB5IF*, *RNF130*, and *TALDO1*) were replicated using an independent blood dataset (GSE59867). Four of the six genes (*COG3*, *CUTC*, *MORF4L1*, and *VTA1*) in the black module showed increased connectivity (GOC) but downregulated expression patterns (Figure 8).

Among the 14 genes in the pink module, 10 (*CBX6*, *ERI3*, *FIBP*, *HGH1*, *PARP6*, *PHGDH*, *POU2F2*, *SPHK2*, *YIF1A*, and *ZNF296*) exhibited LOC in the ACS status (Figure 8). Among the eight genes selected in the dark-red module, five genes (*B4GALT5, CHST15, HAL, SLC22A15*, and *STX3*) were upregulated and exhibited GOC in the ACS status (Figure 8).

In the mouse CVD model, among the nine genes (magenta module) replicated in the human transcriptome dataset (GSE59867), four genes (*ATP6V1A*, *BASP1*, *TKT*, and *ZNF281*) were significantly upregulated in the CVD group in two or more of the four mouse heart datasets. In the brown module, two of the seven genes (*CEBPB* and *TALDO1)* showed increased expression in the mouse heart CVD group.

*CBX6* in the pink module exhibited downregulated expression in most datasets (Figure 9), whereas other genes showed inconsistent differential patterns (i.e., FC between CVD and CN) among the mouse transcriptomic datasets. In case of the dark-red module, *B4GALT5* among the five genes replicated in the human blood dataset provided converged findings (upregulation pattern in the mice heart CVD group) based on the DE analysis (Figure 9).

## 4. Discussion

We identified three obstructive CAD-related modules (black, magenta, and brown) and two ACS-related gene-sets (pink and darkred) in the blood gene expression datasets based on co-expression analysis. These modules are immune- and inflammation-related gene sets. Based on the extreme correlation coefficients (Z-scores ≥ 2), 23 obstructive CAD-related genes (black: *COG3*, *CUTC*, *MAML1*, *MORF4L1*, *NPTN*, and *VTA1*; *magenta*: *ATP6V1A*, *BASP1*, *CHMP2A*, *GCA*, *HNRNPH2*, *HSD17B11*, *NRDC*, *SLC16A3*, *TKT*, and *ZNF281*; *brown*: *ACSL1*, *AGO4*, *CEBPB*, *JPT1*, *RAB5IF*, *RNF130*, and *TALDO1*) and 22 ACS-related genes (pink: *CBX6*, *ERI3*, *FAM50A*, *FIBP*, *GNL1*, *HGH1*, *PARP6*, *PEX26*, *PHGDH*, *POU2F2*, *SNRPB*, *SPHK2*, *YIF1A*, and *ZNF296*; darkred: *ALOX5AP*, *APMAP*, *B4GALT5*, *CHST15*, *HAL*, *LBR*, *SLC22A15*, and *STX3*) were identified.

*ATP6V1A* encodes a component of vacuolar ATPase (*V-ATPase*). *V-ATPase* includes a cytosolic V1 domain and a transmembrane V0 domain, among which the V1 domain contains an ATP catalytic site [38]. *ATP6V1A* is involved in the suppression of acid secretion, growth retardation, trunk deformation, and homeostasis of calcium and sodium ion [39]. Recently, the expression of *ATP6V1A* was reported to be altered in senescent endothelial cells compared with that in nonsenescent cells [40].

*BASP1* interacts with *YY1* and is involved in the regulation of smooth muscle cell proliferation and migration [41]. Several studies have implicated *BASP1* in the pathogenesis of vascular disease. Tian et al. [42] suggested the involvement of *BASP1* in the abdominal aortic aneurysm pathogenesis. Other studies have also reported that *BASP1* contributes to the modulation of angiogenesis [43].

*TKT* encodes transketolase, a thiamine-dependent enzyme that participates in the pentose phosphate pathway (PPP). *Tkt* was recently reported to be dysregulated in activated monocytes [44]. Moreover, the upregulation of nuclear *Tkt* promotes cardiomyocyte apoptosis after myocardial ischemia, leading to cardiac dysfunction [45]. *TALDO1* encodes transaldolase 1, an enzyme involved in the non-oxidative phase of PPP, to restart the oxidative phase [46]. The expression of glycolysis- and PPP-related genes, including *TKT* and *TALDO1*, is reportedly upregulated in patients with CAD-origin chest pain [47].

*ZNF281* is a known human transcription factor (TF) [48]. *ZNF281* participates in controlling cellular stemness and inducing epithelial–mesenchymal transition and cellular differentiation [49,50]. *ZNF281* is also involved in the enhancement of cardiac reprogramming by interacting with *GATA4* and regulating inflammatory signaling [50].

*CEBPB* is a validated human TF [48]. It is a member of the *bHLH* gene family of DNA-binding TFs and participates in cell proliferation and differentiation [51]. *CEBPB* is involved in exercise-induced cardiac hypertrophy and a protective pathway against pathological cardiac remodeling [52]. Recently, *CEBPB* has been reported as a potential neurodegenerative disease-related gene [16].

Genes with high variance across samples in each dataset were initially selected, and approximately 8,000 and 13,000 genes commonly present in two integrated datasets (GSE90074–GSE20681 and GSE34198–GSE60993) were used for the establishment of co-expression networks and modules, respectively. It cannot be denied that the loss of information for background genes had occurred. However, the fact that many genes were still not assigned to a specific module (i.e., the gray module), even though we had conducted co-expression network analysis with only high-quality genes based on variance, might support some of the above preprocessing methods (Appendix A).

Our study presented possible biological or pathophysiological findings. However, our study could not conduct a functional study identifying upregulators affecting the candidate genes we proposed or the cascade pathways affected by them. Future studies, including in vitro and in vivo experiments, are warranted to uncover the complex CVD-related pathogenesis triggered by our candidate blood CVD-related modules or genes. The DE-based previous [16] and DcoE-based present findings can be used for disease prediction panels or prediction models for CVD occurrence or progression [16,53]. Moreover, the CVD gene sets we provided were used as panels to evaluate the effect of intervention (e.g., mesenchymal stem cell therapy [54]) for CVD or a high degree of atherosclerosis.

## 5. Conclusions

In summary, we implemented co-expression network, DE, and DcoE analyses and a mathematical method to construct a gene–gene interaction network to reveal blood CVD-related genes. Moreover, we conducted a replication experiment using human blood and mouse heart CVD datasets independent of the derivation datasets, resulting in several blood CVD-related genes exhibiting robust signatures. The integrative identification framework of the analytical pipeline is a breakthrough in understanding the molecular landscape associated with CVD pathogenesis, specifically in blood. The candidate genes proposed in this study may serve as valuable diagnostic markers or therapeutic targets for CVD.

## Figures and Tables

**Figure 1 cells-11-02867-f001:**
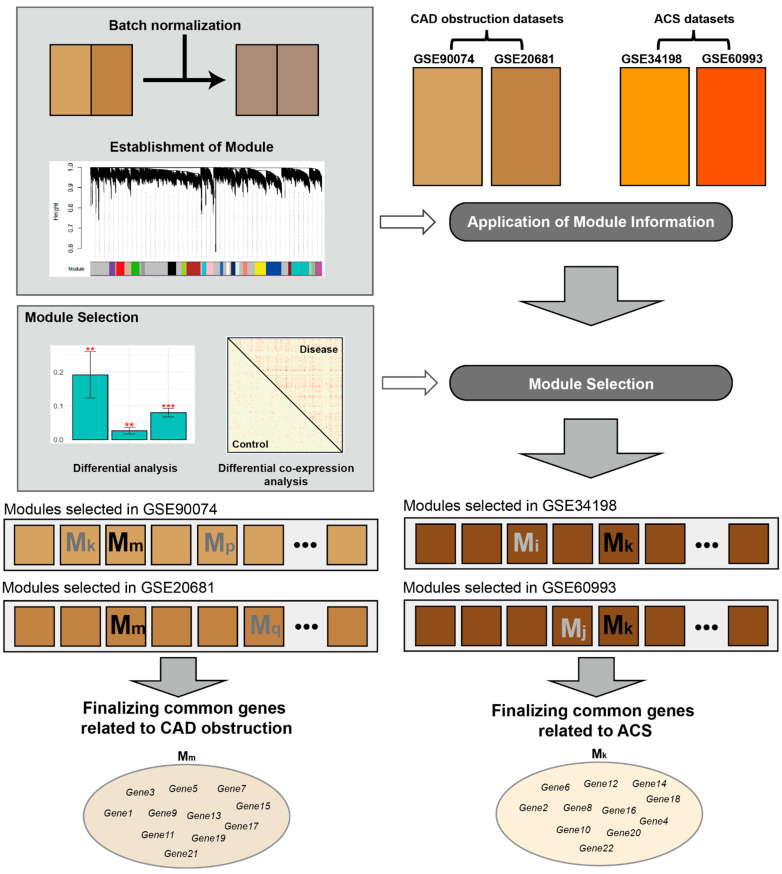
Flow diagram of this study for identifying cardiovascular disease-related modules and genes. GSE90074, GSE20681, GSE34198, and GSE60993 are blood transcriptomic datasets obtained from Gene Expression Omnibus database [19]. Abbreviations: CAD, coronary artery disease; ACS, acute coronary syndrome; GSE, gene expression data series. ** *p* < 0.01, *** *p* < 0.001.

**Figure 2 cells-11-02867-f002:**
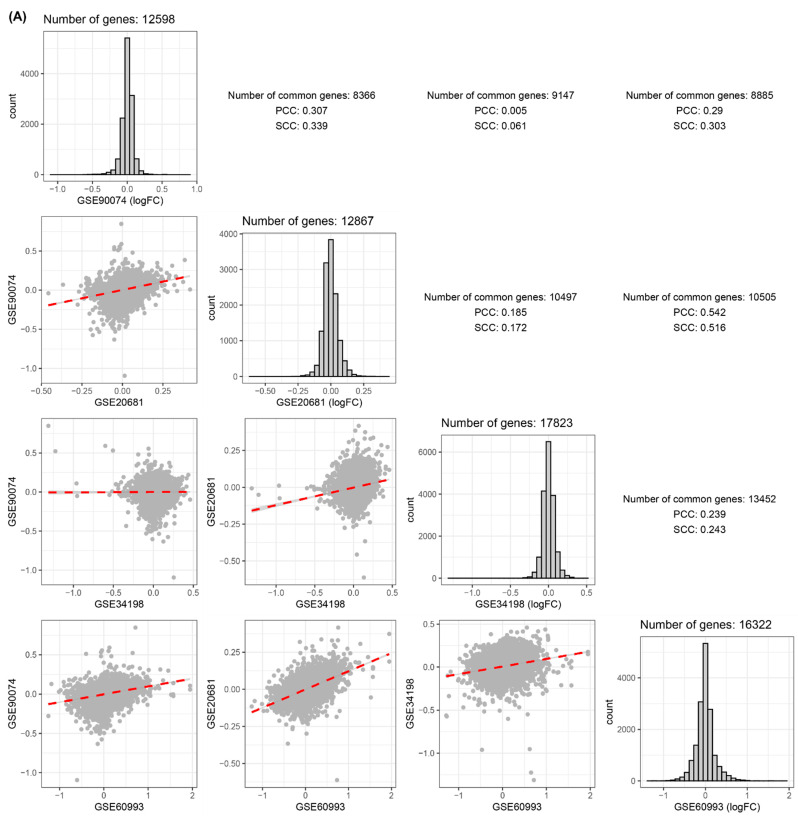
Comparison of disease-related transcriptomic signature among four blood CVD datasets. (**A**) Correlation matrix among four gene expression datasets. Distributions located on the diagonal are expression values for all genes. Each point indicates a gene Y and X axes are logFC between the two statuses described in the form of (A/B). A scatter plot in the xth row and yth column indicates the correlation between xth and yth datasets of diagonal datasets. Its correlation coefficient is described on the xth column and yth row. (**B**) Numbers in upper triangle matrix indicate the number of common genes between each of the four sets of DEGs between two conditions. Their matched *p*-values measured by hypergeometric test are located in lower triangle matrix. (**C**) Biological pathways were based on hypergeometric test described in Materials and Methods. Numbers within rectangles indicate numbers of DEGs or possible DEGs in each pathway. Abbreviations: CVD, cardiovascular disease; FC, fold change; PCC, Pearson’s correlation coefficient; SCC, Spearman’s correlation coefficient; DEG, differentially expressed gene.

**Figure 3 cells-11-02867-f003:**
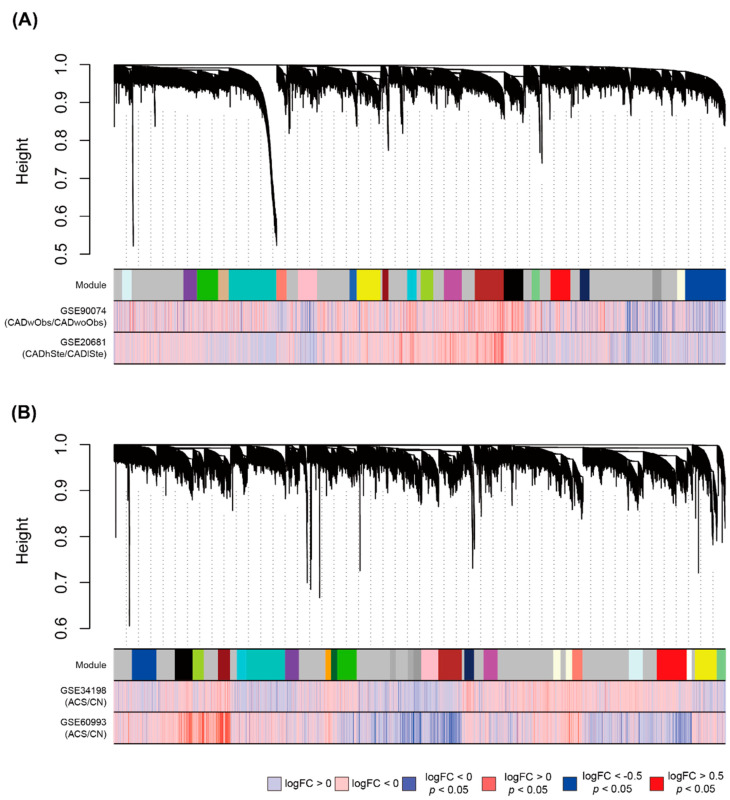
Establishment of modules via weighted gene co-expression network analysis (WGNCA). (**A**) Use of GSE90074 and GSE20681 for construction of CAD obstruction-related modules and (**B**) use of GSE34198 and GSE60993 for construction of ACS-related gene sets. Modules include genes with high correlation based on biweight midcorrelation and are depicted with different colored planes. Each plane consists of numerous lines of the same color, with a single line representing a gene. Color bars in the lower part indicate fold change (FC) values (logFC) between two conditions calculated by the limma package. Height indicates “1 − (topological overlap)”, described by the network dendrogram. Abbreviations: FC, fold change; CAD, coronary artery disease; CADwObs, CAD with obstruction; CADwoObs, CAD without obstruction; ACS, acute coronary syndrome; CN: matched control.

**Figure 4 cells-11-02867-f004:**
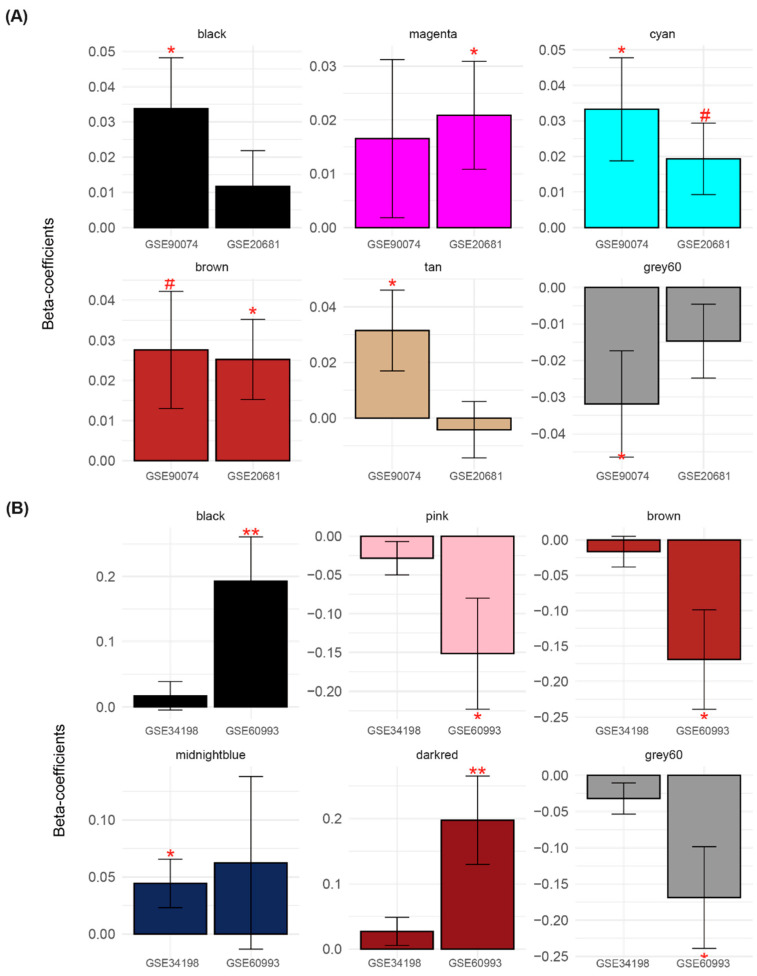
Selection of obstructive CAD- and ACS-related modules based on differential expression analysis. Beta-coefficients were measured from linear regression by setting MEs (**A**) and disease status (**B**) (i.e., obstructive CAD and ACS vs. matched control) as dependent and independent variables, respectively. **, *, # denote *p* < 0.01, *p* < 0.05, and *p* < 0.1, respectively. ME, module eigengene; CAD, coronary artery disease; ACS, acute coronary syndrome.

**Figure 5 cells-11-02867-f005:**
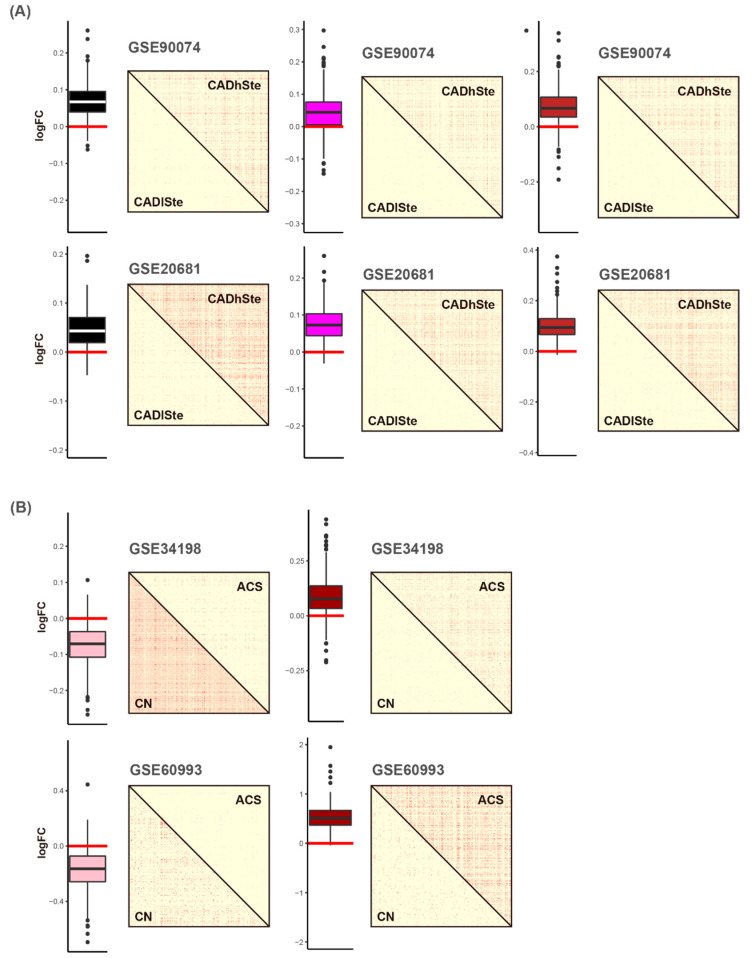
Selection of CAD obstruction (**A**) and ACS-related (**B**) modules based on differential co-expression analysis. Fold change (FC) values (logFC) between two conditions were calculated using the limma package. Adjacency matrices are described as six matrix plots. A matrix plot consists of two conditions: disease status including CAD obstruction and ACS is the upper triangular matrix, and matched control status is the lower triangular matrix. Abbreviations: FC, fold change; CAD, coronary artery disease; CADhSte, CAD with high stenosis; CADlSte, CAD with low stenosis; ACS, acute coronary syndrome; CN, matched control.

**Figure 6 cells-11-02867-f006:**
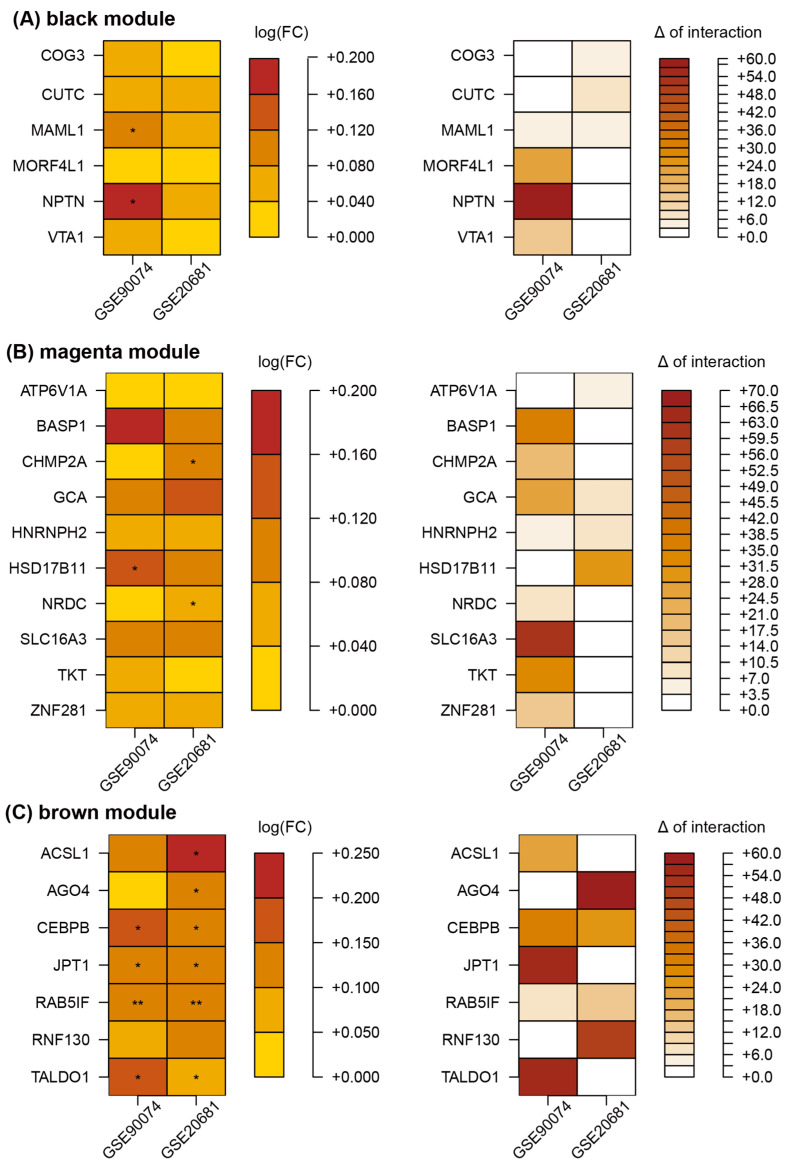
Representative obstructive CAD-related genes in the black (**A**), magenta (**B**), and brown (**C**) modules. FC values (logFC) between two conditions were calculated using the limma package. Delta (Δ) of interaction indicates the difference between the number of interactions in CVD and matched control groups. **, and * denote *p* < 0.01, and *p* < 0.05, respectively. Abbreviations: CAD, coronary artery disease; FC, fold change.

**Figure 7 cells-11-02867-f007:**
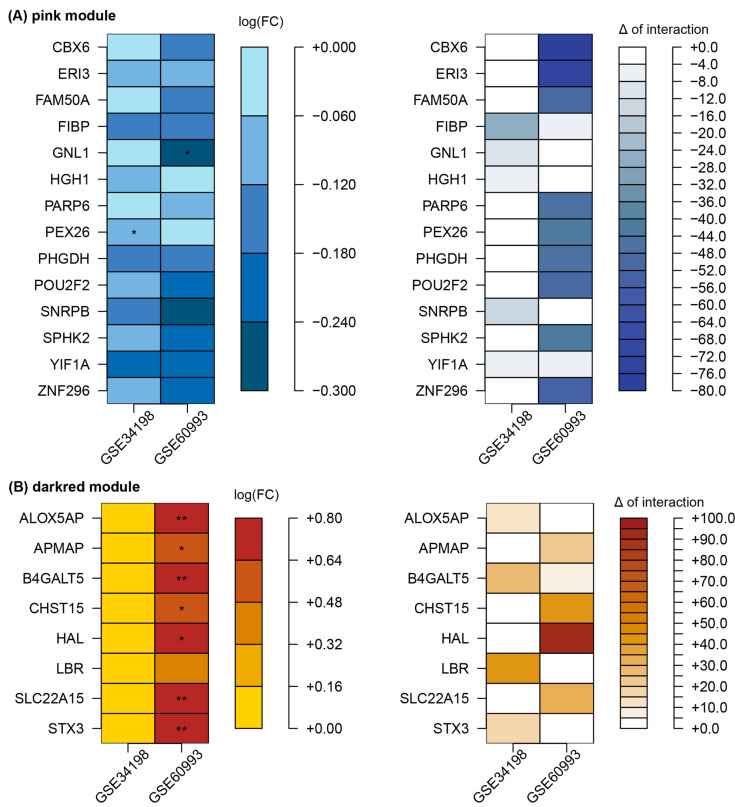
Representative ACS-related genes in the pink (**A**) and dark-red (**B**) modules. FC values (logFC) between two conditions were calculated using the limma package. Delta (Δ) of interaction indicates the difference between the number of interactions in ACS and matched control groups. **, and * denote *p* < 0.01, and *p* < 0.05, respectively. Abbreviations: CAD, coronary artery disease; FC, fold change.

**Figure 8 cells-11-02867-f008:**
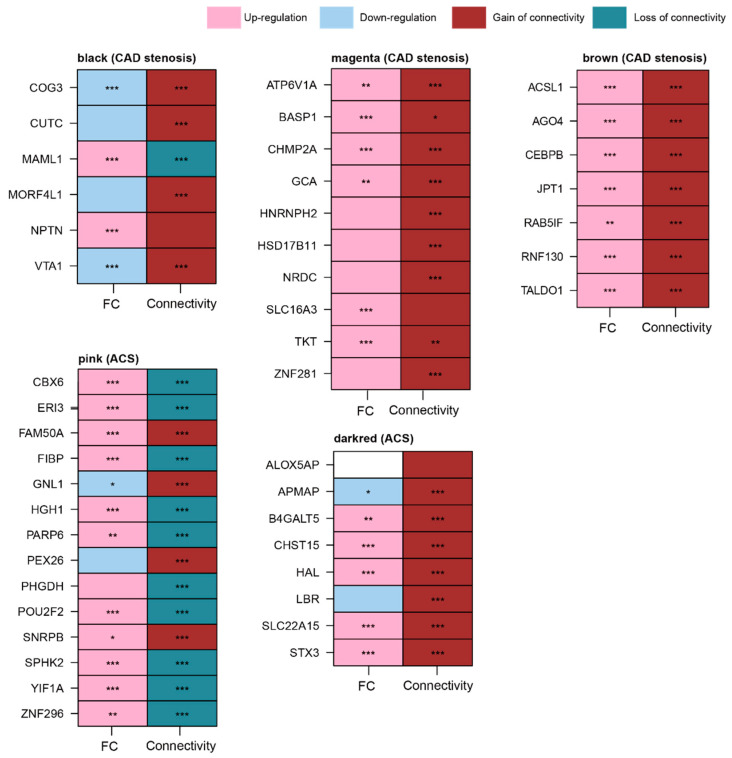
Replication analyses from independent human blood CVD datasets for genes in five modules (black, magenta, brown, pink, and dark red). FC values (logFC) between two conditions were calculated using the limma package. Connectivity was measured based on biweight midcorrelation. In logFC columns, ***, **, and * denote *p.adj* < 0.001, *p.adj* < 0.01, and *p.adj* < 0.05, respectively, and these values were measured by the limma method. In connectivity columns, ***, **, and * denote *p* < 0.001, *p* < 0.01, and *p* < 0.05, respectively, and these values were measured by paired *t*-test between ACS and matched CN correlation matrices. Abbreviations: CVD, cardiovascular disease; CAD, coronary artery disease; ACS, acute coronary syndrome; FC, fold change; *p.adj*, *p*-value adjusted by false discovery rate.

**Figure 9 cells-11-02867-f009:**
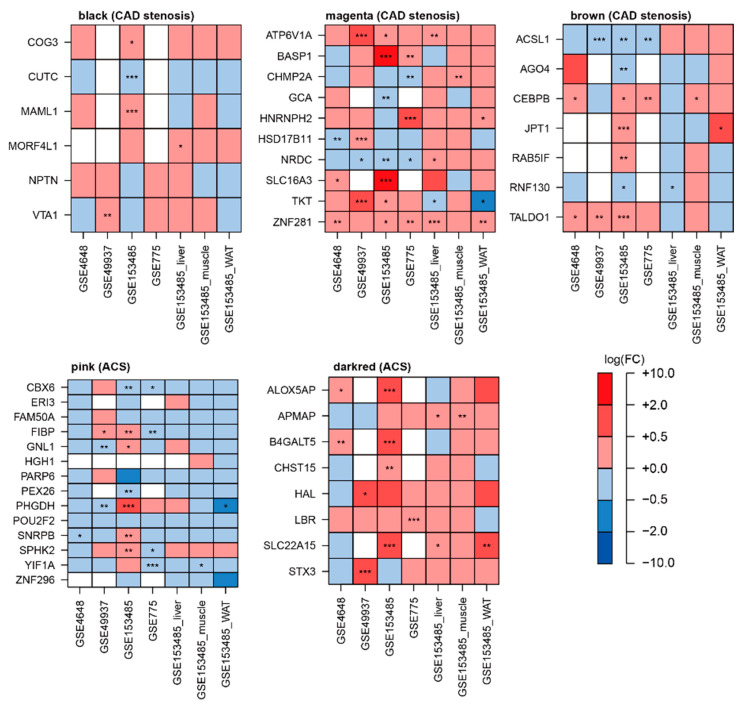
Replication analyses from independent mice CVD datasets for genes in five modules (black, magenta, brown, pink, and dark red). FC values (logFC) between two conditions were calculated using the limma package. The empty rectangles colored in white are gene expression levels that are not included in the animal data. ***, **, *, and # denote *p.adj* < 0.001, *p.adj* < 0.01, *p.adj* < 0.05, and *p* < 0.05, respectively. Black, magenta, and brown are obtained from CAD stenosis-related datasets and pink and darkred modules were made from ACS-related datasets. Abbreviations: CVD, cardiovascular disease; FC, fold change; *p.adj*, *p*-value adjusted by false discovery rate.

## Data Availability

Gene expression datasets are publicly available (GEO, https://www.ncbi.nlm.nih.gov/geo, accessed on 9 September 2022).

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
