# Peer review of "Identification of Cardiovascular Disease-Related Genes Based on the Co-Expression Network Analysis of Genome-Wide Blood Transcriptome"

_cells, 2022, doi:10.3390/cells11182867_

Round 1
Reviewer 1 Report
Please refer to the attached file.

Author Response
The independent PDF file including answers to reviewer's comments was uploaded.

Reviewer 2 Report
In this manuscript, Lee et al describe their approach to identifying genes related to cardiovascular diseases (CVDs) using the co-expression network analysis of genome-wide 3 blood transcriptome. This is an interesting topic as CVDs in one of the major causes of death globally. Identifying CVD-related genes may contribute to the prevention and/or treatment of the disease. The manuscript is well written and the results/main findings are extensively discussed. However, I have some minor suggestions that can be incorporated to strengthen the review as follows:
1. The introduction should be more informative and the aim of this work should be clearly stated in the last paragraph (LL 63-72)
2. LL 244-245. Please provide the name of the three databases
3. The authors should provide some suggestions for future studies.
Author Response

(The authors gave the same response as above.)

Round 2
Reviewer 1 Report
The authors have addressed my previous comments.